# *Eucommia ulmoides* Flavones as Potential Alternatives to Antibiotic Growth Promoters in a Low-Protein Diet Improve Growth Performance and Intestinal Health in Weaning Piglets

**DOI:** 10.3390/ani10111998

**Published:** 2020-10-30

**Authors:** Daixiu Yuan, Jing Wang, Dingfu Xiao, Jiefeng Li, Yanhong Liu, Bie Tan, Yulong Yin

**Affiliations:** 1Department of Medicine, Jishou University, Jishou 416000, China; yuandaixiu123@126.com; 2Department of Animal Science, Hunan Agricultural University, Changsha 410000, China; xiaodingfu2001@163.com (D.X.); bietan@isa.ac.cn (B.T.); 3Laboratory of Animal Nutritional Physiology and Metabolic Processes, National Engineering Laboratory for Pollution Control and Waste Utilization in Livestock and Poultry Production, Key Laboratory of Agroecological Processes in Subtropical Region, Institute of Subtropical Agriculture, Chinese Academy of Sciences, Changsha 410125, China; lijiefeng1997@126.com; 4Department of Animal Science, University of California, Davis, CA 95616, USA; yahliu@ucdavis.edu

**Keywords:** antibiotic alternatives, *Eucommia ulmoides* flavones, growth performance, intestinal barrier, weaning piglets

## Abstract

**Simple Summary:**

Antibiotic resistance is a growing threat to the effective treatment of bacterial infections in both humans and animals, making the treatment of patients and livestock more difficult or even impossible. *Eucommia ulmoides* flavones (EUF), which were extracted from *Eucommia ulmoides* leaf, have been shown strong antioxidant properties and the inhibition of pro-inflammation in our previous studies. We found that EUF could promote growth performance, improve intestinal health, and reduce colonization of coliform bacteria and diarrhea index in weanling piglets; and this suggests that EUF may be a potential alternative to in-feed antibiotic growth promoter in pig husbandry.

**Abstract:**

*Eucommia ulmoides* flavones (EUF) have been demonstrated to attenuate the inflammation and oxidative stress of piglets. This study aimed to test whether EUF could be used as an alternative antibiotic growth promoter to support growth performance and maintain intestinal health in weanling piglets. Weaned piglets (*n* = 480) were assigned into three groups and fed with a low-protein basal diet (NC), or supplementation with antibiotics (PC) or 0.01% EUF (EUF). Blood, intestinal contents, and intestine were collected on days 15 and 35 after weaning. The results showed the PC and EUF supplementations increased (*p* < 0.05) body weight on day 35, average daily gain and gain: feed ratio from day 15 to day 35 and day 0 to day 35, whereas decreased (*p* < 0.05) the diarrhea index of weanling piglets. EUF treatment increased (*p* < 0.05) jejunal villus height: crypt depth ratio, jejunal and ileal villus height, and population of ileal lactic acid bacteria on day 15 but decreased (*p* < 0.05) the population of ileal coliform bacteria on day 15 and day 35. These findings indicated the EUF, as the potential alternative to in-feed antibiotic growth promoter, could improve growth performance and intestinal morphology, and decrease colonization of coliform bacteria and diarrhea index in weanling piglets.

## 1. Introduction

Antibiotic has been used for decades to decrease pathogen infection, but increasing scientific evidence was gathered on a relationship between medication and pathogenic resistance [1,2]. Indeed, announcement No. 194 of the Ministry of Agriculture and Rural Affairs of the People’s Republic of China stipulates that medicated feed additives will be prohibited from being used in animal feed in 2020 [3]. Antibiotics-free diets have become a necessity in the livestock and poultry industry. The withdrawal of antibiotic use will lead to lower quality and yield of animals or death, more serious disease outbreaks, and then result in greater use of antibiotics for therapeutic purposes [4]. Therefore, new antibiotic alternative strategies are needed to guarantee animal health and yield growth. It has been well documented that some antibiotic alternatives and feed additives, such as plant extracts, organic acids, microecologics, and antimicrobial peptides, could promote the animal development and enhance the intestinal health [5,6,7,8].

Of the various alternative, plant extract is one of the most readily available and safe choice that being investigated [7,8]. *Eucommia ulmoides* (Chinese: Duzhong) is a species of small tree native to China, which belongs to the monotypic family *Eucommiaceae* [9]. *Eucommia ulmoides* is one of the fifty fundamental herbs used in Chinese herbology, and it contains various chemical compounds such as lignans, iridoids, phenolics, steroids, flavonoids, and other compounds [9]. Because of the high value of *Eucommia ulmoides*’ bark in herbology, *Eucommia ulmoides* is widely cultivated in China. In addition to the *Eucommia ulmoides’* bark, other parts of this plant also exist bioactivities [10]. The *Eucommia ulmoides* flavones (EUF), which are bioactive phytochemicals derived from the leaves of *Eucommia ulmoides*, have been demonstrated to improve the antioxidative function by reacting with free radicals [11]. It has been reported that EUF could improve microbial balance and reduce the response of inflammation [12,13], which may help to enhance the host defense and physical barrier function. Our previous study has also showed that EUF alleviate the oxidative stress induced by diquat in piglets by reducing the growth performance impairment, pro-inflammatory cytokines secretion and intestinal barrier dysfunction [14,15]. Meanwhile, we recently report the EUF could modulate the NF-E2-related factor 2 (Nrf2) signaling pathway in the intestine to mitigate the oxidative stress of piglets [16]. The Nrf2 pathway not only involves in antioxidant by regulating the mRNA levels of antioxidant enzymes, but also enhances the intestinal barrier integrity through increasing the expression of tight junction proteins [17], which may suggest that EUF has positive impact on intestinal barrier integrity. The improved functional gut immunity and integrity are vital to reduce the permeability for viable pathogens and pathogen colonization in the gut [17]. These benefit effects may enable it to be an effective antibiotic alternative to promote animal growth in animal husbandry.

Therefore, the present study was conducted to investigate the effect of EUF as antibiotic alternative in a low-protein diet on growth performance and intestinal health of weaning piglets. The low-protein diet was used in this study to promote gut health and maintain the normal digestion and absorption capacities of enterocytes without impairing the growth performance of piglets [18]. Growth performance, serum biochemical parameters, intestinal morphology and microbiota composition were monitored so as to provide the scientific basis for the application of EUF in antibiotics-free diets in swine production.

## 2. Materials and Methods

The animal experiments were approved by the Institutional Animal Care and Use Committee of Hunan Agricultural University, Hunan, China. The animal protocol was approved by Institutional Animal Care and Use Committee (IACUC No. 20190056).

### 2.1. Animal Protocol and Dietary Treatment

A total of 480 piglets (Duroc × Landrace × Large Yorkshire) weaned at 25 days of age were randomly assigned to 3 groups (8 pens per group and 20 piglets per pen) as follows: (1) negative control (NC), low-protein basal diet no antibiotics included; (2) positive control (PC), low-protein basal diet + antibiotics (75 mg/kg quinocetone, 20 mg/kg virginomycin and 50 mg/kg aureomycin); (3) EUF treatment (EUF), low-protein basal diet + 0.01% EUF. The low-protein basal diet was formulated in two phases (the first day of weaning was designated as day 0; phase 1, days 0–15; phase 2, days 15–35) according to the nutrient requirements for weanling piglets (NRC, 2012) and the previous studies [19] (Table 1). *Eucommia ulmoides* came from Xiangxi (Hunan, China), and the powder contained 83.61% total flavones was prepared at the department of medicine, Jishou University (Jishou, Hunan, China), which has been used in the previous study by Yuan et al. [14]. 

The piglets were housed in an environmentally controlled nursery with hard plastic slatted flooring, and had free access to drinking water. Piglets were fed their respective diets ad libitum for a 35-day period. On the morning at days 15 and 35, 24 piglets (1 piglet per pen) were randomly selected and blood samples were obtained aseptically from the jugular vein at 2 h after feeding. Serum samples were obtained by centrifugation at 2000× *g* for 10 min at 4 °C and then immediately stored at −80 °C for further analysis. Piglets were anesthetized with sodium pentobarbital and sacrificed by jugular puncture. About 2 cm segments of the jejunum and ileum were cut and fixed in 4% formaldehyde for observation of the morphology of intestinal mucosa. Ileal and colonic contents were collected for bacteria counting and DNA extraction.

### 2.2. Growth Performance and Diarrhea Index

Body weight was obtained on days 0, 15, and 35. The average daily gain was calculated as the terminal body weight minus the initial body weight and then divided into the number of days. Both the amount of the diet offered and the leftovers were recorded daily to obtain the daily feed intake of each pen. The daily feed intake per animal was calculated as daily feed intake per pen divided by the number of piglets per pen. The gain:feed ratio was calculated as the average daily gain divided by the average daily feed intake. The number of pigs with diarrhea was recorded 3 times a day at 8:00, 13:00 and 18:00 for a 35-day period. A piglet with diarrhea case (≥1) in 3 observations would be recorded as one piglet diarrhea. Diarrhea index (%) was calculated as the number of diarrhea piglets × diarrhea days/the total number of piglets × experiment days.

### 2.3. Intestinal Morphology Evaluation

The jejunal and ileal morphology were analyzed by using hematoxylin–eosin staining according to Wang et al. [20]. The segments of the jejunum and ileum fixed in 4% formaldehyde were used to determine morphology using hematoxylin–eosin staining. After dehydration, embedding, sectioning, and staining, villous height and crypt depth were measured with computer-assisted microscopy (Micrometrics TM; Nikon ECLIPSE E200, Tokyo, Japan). Villous height and crypt depth, counts were measured by Image-Pro Plus software, Version 6.0 on images at 40-fold magnification in five randomly selected fields. The villous height was measured from top of villous to the crypt opening, and crypt depth was measured from the base of the crypt to the level of crypt opening.

### 2.4. Serum Biochemical Parameters Determination

Total protein (TP), albumin (ALB), blood urea nitrogen (BUN), glucose (GLU), alanine aminotransferase (ALT), aspartate aminotransferase (AST), alkaline phosphatase (ALP), immunoglobulin G (IgG), and IgM in the serum were measured, using Biochemical Analytical Instrument (Cobas c311, F. Hoffmann-La Roche Ltd., Basel, Switzerland) and commercial kits (F. Hoffmann-La Roche Ltd., Basel, Switzerland).

### 2.5. Microbiota Composition Analysis

Bacteria counting was performed according to the previous studies [7,21]. First, 0.2 g of ileal and colonic contents was weighed and immediately diluted with 9 mL of 0.9% sterilizing saline and homogenized. Then, 10-fold dilutions of homogenate were performed (ranging from 10^−1^ to 10^−8^) and then cultivated onto MacConkey Agar Medium, for the enumeration of *Escherichia coli*, and GM17 Medium for the enumeration of lactic acid bacteria. The GM17 medium were then incubated for 48 h at 30 °C under anaerobic conditions, while the MacConkey agar plates were incubated for 24 h at 37 °C. The coliform bacteria and lactic acid bacteria colonies were counted immediately after removal from the incubator. Values were reported as log10 colony-forming units per gram.

DNA was extracted from ileal and colonic contents with the Tiangen stool mini kit (TianGen) according to the instructions of the manufacturer. DNA concentration was determined by spectrophotometry (Nanodrop). The DNA obtained from the intestinal luminal content was used as the template to analyze intestinal bacteria by qRT-PCR. Primers (*Lactobacillus* spp. (F) 5′-CACCGCTACACATGGAG-3′ (R) 5′-TGGAAGATTCCCTACTGCT-3′, *Escherichia coli* (F) 5′-CATGCCGCGTGTATGAAGAA-3′ (R) 5′-TTTGCTCATTGACGTTACCCG-3′, and total bacteria (F) 5′-ACTCCTACGGGAGGCAGCAG-3′ (R) 5′-ATTACCGCGGCTGCTGG-3′) were synthesized according to the previous study [22]. Relative expression of genes in the treatment group was normalized to the values for the NC.

### 2.6. Statistical Analysis

The goal of the current study is to determine whether EUF could be an alternative to antibiotic growth promoter, and the developmental changes of growth performance and gut health from day 0 to day 35 post-weaning are not main purpose. In the present study, the treatment effects (NC vs. PC vs. EUF) on variables were performed by SPSS v.23 software package (SPSS Inc. Chicago, IL, USA). Data are expressed as mean ± SEM. The differences among treatments were evaluated by using one-way ANOVA with Tukey’s test, following by the Kruskal–Wallis test when data were not normally distributed. Probability values of *p* < 0.05 were taken to indicate statistical significance.

## 3. Results

### 3.1. Growth Performance and Diarrhea Index

The average daily gain, average daily feed intake, and gain: feed ratio were showed in Figure 1. The body weight on days 0 and 15 were not different among all treatments (*p* > 0.05). The body weight of piglets in PC and EUF treatments were 18.4% and 21.0% higher (*p* < 0.05) than those in the NC treatment on day 35, respectively. Compared with the piglets of NC treatment, PC and EUF treatments increased average daily gain and gain: feed ratio from days 15 to 35 and days 0 to 35 (*p* < 0.05). However, there was no difference in average daily feed intake of all experimental periods among all treatments (*p* > 0.05). The diarrhea index in the piglets of PC and EUF treatments from days 0 to 15 and days 0 to 35 were significantly reduced by 64.4% (PC vs. NC, days 0–15), 60.7% (PC vs. NC, days 0–35), and 64.3% (EUF vs. NC, days 0–15), 59.3% (EUF vs. NC, days 0–15), compared with NC treatment (*p* < 0.05) (Figure 2). The PC and EUF groups were not different at the variables mentioned above (*p* > 0.05).

### 3.2. Serum Biochemical Parameters

All the serum biochemical parameters were in physiological range [23], and the values at day 15 and day 35 were comparable. Dietary supplementation with EUF increased (*p* < 0.05) the serum concentrations of TP (EUF vs. NC, 23.88%) and IgG (EUF vs. NC, 71.32%), as well as ALT (EUF vs. NC, 46.29%) activity on day 15 and serum TP (EUF vs. NC, 17.06%) content on day 35, compared to NC treatment. PC treatment had trends to increase the serum ALT activity, IgG level on day 15 and TP content on day 35 compared to NC group, but the differences were not significant. In comparison to PC treatment, EUF addition enhanced (*p* < 0.05) serum TP content and ALP activity on day 15 by 19.14% and 19.22%, respectively. There was no difference (*p* > 0.05) in other determined serum biochemical parameters on day 15 or 35 (Table 2).

### 3.3. Microbiota Composition in Ileum and Colon

In the ileum, the values of lactic acid bacteria and coliform bacteria numbers on days 15 and 35 were similar. EUF addition increased the number of lactic acid bacteria on day 15 but decreased the number of coliform bacteria on days 15 and 35, compared with NC treatment (*p* < 0.05). PC had a trend to increase the number of lactic acid bacteria, but the difference was not significant. Moreover, no significant difference was observed in the numbers of lactic acid bacteria and coliform bacteria between PC and EUF treatments on days 15 and 35 (*p* > 0.05) (Table 3). For the relative abundance of selected microbiota, both PC and EUF treatments significantly decreased the population of *Escherichia coli* on day 15 (PC vs. NC, 32%; EUF vs. NC, 43%) and 35 (PC vs. NC, 37%; EUF vs. NC, 25%) but increased the population of *Lactobacillus* spp. on day 15 (PC vs. NC, 46%; EUF vs. NC, 50%) in the ileum, compared to those of the NC treatment (*p* < 0.05) (Figure 3). There was no difference in determined microbiota in the colon among three treatments on days 15 and 35 (*p* > 0.05).

### 3.4. Intestinal Morphology

The representative images and the results of intestinal morphology were presented in Table 4 and Appendix A
Appendix A, respectively. Dietary supplementation with EUF and PC increased (*p* < 0.05) the jejunal and ileal villus height on day 15, compared to the NC treatment. EUF administration increased (*p* < 0.05) the ratio of villus height to crypt depth of jejunum on day 35 by 15.88%, in comparison with NC group. No difference was observed on villus height and crypt depth, as well as the villus height: crypt depth ratio in jejunum and ileum between PC and EUF treatments on days 15 and 35 (*p* > 0.05) (Table 4).

## 4. Discussion

A previous study has shown the beneficial effect of *Eucommia ulmoides* leaf extracts on the intestinal barrier function and digestive enzyme activities compared to antibiotics in piglets fed by a normal protein level diet (CP = 20.72%) [24]. The present study demonstrated that EUF supplementation in a low-protein and antibiotic-free diet has a comparable growth-promoting effect with commercial antibiotics, such as the improved growth performance and intestinal morphology, lower colonization of coliform bacteria and diarrhea index in comparison with the antibiotic-free control group. A low-protein diet was used as basal diet, which could improve intestinal health but do not affect the growth performance. A low-protein diet not only decreasing the cost of feed but also alleviating the nutritional burden of excess protein loading, reducing the protein fermentation that damages the intestinal barrier function [25]. The practice of using a low-protein diet in the livestock and poultry industry is discouraged and becoming increasingly prevalent in China. In this study low-protein diet was used as a basal diet and recommended to work with EUF, as an antibiotic alternative nutritional strategy, to profitably improve the growth performance and gut health without antimicrobial growth promoters.

Ideal antibiotic alternatives should have the same beneficial effects of antibiotic growth promoters, ensuring optimum animal performance and nutrient availability [8]. Similar increases in body weight, average daily gain and feed efficacy between the piglets of antibiotic positive control and EUF treatment were obtained in our present study. In addition, serum TP level and ALP activity could reflect the body protein metabolic stasis [26] and the alteration of soft tissue membrane permeability [27,28], respectively. Although these parameters’ values were in physiological ranges, EUF supplementation promoted serum TP content and ALP activity compared to antibiotic positive control, which in a certain extent showing the increased nutrient efficiency and development of piglets and their organs [29]. The function of ALT is to convert alanine into pyruvate, which is an important intermediate in cellular energy production. Serum ALT activity is considered as the most frequently relied upon indicator of hepatotoxic effects of drugs [30]. In the current study, EUF increased the serum ALT activities on day 15, while the value was in physiological ranges, which indicated EUF has no hepatotoxic effect but suggested EUF may involve in the gluconeogenesis regulation of young piglets. It is important to develop an alternative strategy to stimulate innate immune response and limit the infections in livestock, and subsequently decrease the use of antibiotics [31]. IgG is a feature of immune cell maturation and plays a critical role in defensing against infection via the direct neutralization of toxins and microbes [32]. In our study, serum IgG concentrations were enhanced by dietary supplementation with EUF, which is also consistent with previous published report that EUF exerts immunomodulatory activities by modifying the production of cytokines in vivo [14] and regulating the NF-κB pathway in vitro [33]. Although the IgG levels in EUF group and antibiotic positive control group were not significantly different, the antibiotic growth promoters did not increase the IgG concentration in piglets compared to antibiotic-free negative control group as EUF did. Accordingly, EUF may have great potential as alternative antibiotics to improve immunity and protect piglets from pathogen infection by mediating the NF-κB pathway and partly depending on its antioxidative capacity.

The proposed mechanism of promoting growth effects of a practical alternative may be involved in modulating the gut microbiome [7,8]. Weaning stress induces the population of pathogenic *Escherichia coli* to proliferate to exceed those of other bacterial populations, which is associated with many diseases after weaning [34]. Removing antibiotics from the diet will inevitably lead to a further increase in the number of microorganisms [35]. In the current study, antibiotic-free dietary supplementation with EUF significantly increased the population of lactic acid bacteria and decreased coliform bacteria whereas the antibiotic treatment only reduced the coliform bacteria content but did not affect the lactic acid bacteria content, which may indicate the activity of prebiotic of EUF compared to commercial antibiotics. Lactic acid bacteria are one of the most commonly used probiotics in livestock, and predominant at the early stage of pig gut microflora construction [36]. Increased lactic acid bacteria can reduce fecal pH and ammonia nitrogen levels [37], as well as prevent colonization of pathogenic organism colonization; therefore, it can improve the natural microbiome and gut health [38]. Lots of bioactive antimicrobial chemical forms, including phenolic acids, quinones, flavonoids, tannins, and alkaloids, have been identified and been used in animal nutrition [7,8]. However, due to their complex compositions and the potential for multiple sites of action, the mechanisms of antibacterial activity are not fully understood. One of the mechanisms of inhibitory action is involved that hydroxyl groups in phenolic compounds interact with the cell membrane of bacteria to disrupt membrane structures and cause the leakage of cellular components [39].

Recent studies have explored that antibiotic exposure early in life has long-term consequences on intestinal homeostasis and epithelial barrier function [40]. In addition to antibacterial activity, the effect of EUF on the intestinal barrier function was investigated in the present study. Although we did not detect the tight junction protein expression and intestinal permeability, the significantly increased villus height: The crypt depth ratio in jejunum and villus height in jejunal and ileal mucosa were observed in EUF supplemented group. In comparison to the antibiotic positive control group, EUF did not show a remarkable advantage in the intestinal morphology, which may due to the beneficial effect of the low-protein diet on the intestine and fewer challenges compared to the negative control group. In our previous study, EUF improves the morphology structure and barrier function of intestine in piglets challenged by diquat exhibiting higher intestinal villus height and lower serum concentrations of D-lactate and diamine oxidase [14]. The intestinal barrier is the first line of defense against pathogen attachment to and invasion of epithelial cells [41]. The effect of EUF on intestinal barrier function may benefit its ability to enhance host defense against microbial infections [8,42].

It should be mentioned that the efficacy of EUF supplementation on piglets was inconsistent between day 15 and day 35. The post-weaning diarrhea, the dynamic restoration of intestinal barrier, and the functional rearrangement of bacterial community in piglets mainly occurred during the first two weeks after weaning [37]; thereafter, the microbiota population and intestinal barrier are relatively stable and maturational [43]. Our results showed that the dietary EUF intervention has a more optimal impact on piglet health and growth in days 0–15 than days 15–35 after weaning. It might be explained that the intestine of piglet during days 0–15 post-weaning is more susceptible to the infections [20,44] and has more urgent needs to restore the intestinal homeostasis and bacterial balance, while the mechanism remains unknown.

## 5. Conclusions

In summary, flavones extracted from *Eucommia ulmoides* leaf have shown the positive impact on growth, immunity, and microbial homeostasis in weanling piglets fed a low-protein diet. In addition to the improved growth performance, mature intestinal barrier, and higher serum IgG level, as well as lower colonization of coliform bacteria and diarrhea index compared to the antibiotic-free negative control, the EUF shows advantages in raising serum total protein and IgG levels, increasing the population of lactic acid bacteria compared to antibiotic growth promoters. These findings can contribute to exploration of EUF as potential alternative to in-feed antibiotic growth promoter to against the microbe infection in swine production although further studies are needed to further explain the mechanism.

## Figures and Tables

**Figure 1 animals-10-01998-f001:**
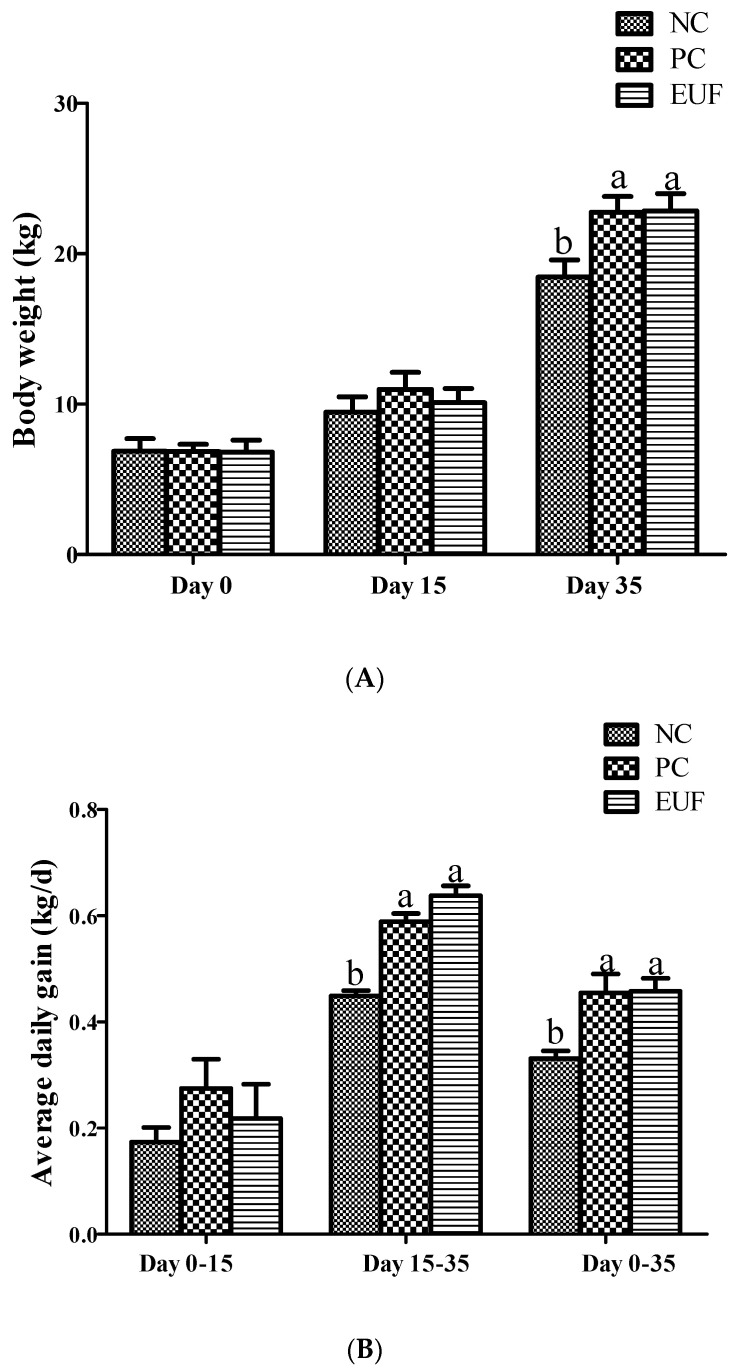
Growth performance: (**A**) body weight; (**B**) average daily gain; (**C**) average daily feed intake; (**D**) gain Z: feed ration; NC = negative control, low-protein basal diet no antibiotics included; PC = positive control, low-protein basal diet + antibiotics (75 mg/kg quinocetone, 20 mg/kg virginomycin, and 50 mg/kg aureomycin); EUF = *Eucommia ulmoides* flavones, low-protein basal diet + 0.01% EUF. Values are the mean ± SEM, *n* = 8 per treatment group. ^a,b^ Values with different letters are significantly different (*p* < 0.05).

**Figure 2 animals-10-01998-f002:**
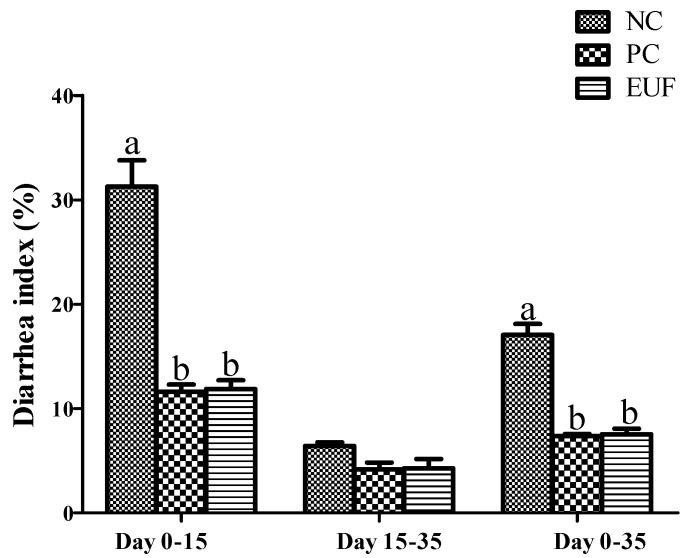
Diarrhea index: NC = negative control, low-protein basal diet no antibiotics included; PC = positive control, low-protein basal diet + antibiotics (75 mg/kg quinocetone, 20 mg/kg virginomycin, and 50 mg/kg aureomycin); EUF = *Eucommia ulmoides* flavones, low-protein basal diet + 0.01% EUF. Values are the mean ± SEM, *n* = 8 per treatment group. ^a,b^ Values with different letters are significantly different (*p* < 0.05).

**Figure 3 animals-10-01998-f003:**
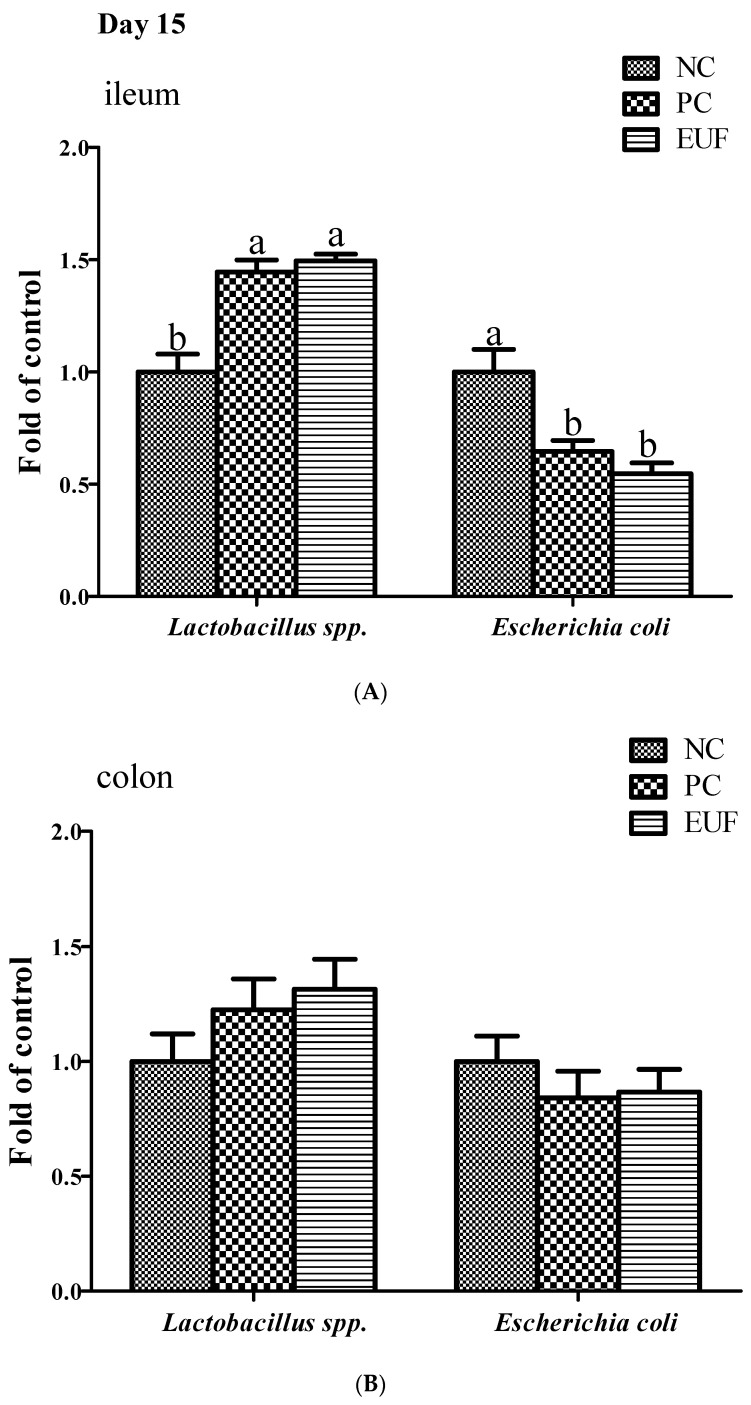
Relative abundance of *Lactobacillus* spp. and *Escherichia coli* in the ileum (**A**,**B**) and colon (**C**,**D**) of piglets. NC = negative control, low-protein basal diet no antibiotics included; PC = positive control, low-protein basal diet + antibiotics (75 mg/kg quinocetone, 20 mg/kg virginomycin and 50 mg/kg aureomycin); EUF = *Eucommia ulmoides* flavones, low-protein basal diet + 0.01% EUF. Values are the mean ± SEM, *n* = 8 per treatment group. ^a,b^ Values with different letters are significantly different (*p* < 0.05).

**Table 1 animals-10-01998-t001:** Composition of basal diets (as-fed basis).

Ingredients, %	Phase 1	Phase 2
Corn	57	60
Expended maize	5	5
Soybean meal (43% CP)	22	22
Rice bran meal	5	5
Broken rice	5	5
Fish meal	2	/
Sucrose	1	/
Calcium lactate	0.3	0.3
Calcium hydrogen phosphate	1	1
Limestone powder	0.1	0.1
Premix ^1^	1	1
98% lysine	0.4	0.4
Threonine	0.1	0.1
Chemical composition ^2^		
Dry matter	87.86	88.58
Crude protein	17.4	16.3
Calculated DE, kcal/kg	3466	3420
Lysine	0.79	0.71
Calcium	0.68	0.63
Total phosphorus	0.53	0.46

^1^ Providing the following amounts of vitamins and minerals per kilogram on an as-fed basis: Zn (ZnO), 50 mg; Cu (CuSO_4_), 20 mg; Mn (MnO), 55 mg; Fe (FeSO_4_), 100 mg; I (KI), 1 mg; Co (CoSO_4_), 2 mg; Se (Na_2_SeO_3_), 0.3 mg; vitamin A, 8,255 IU; vitamin D3, 2000 IU; vitamin E, 40 IU; vitamin B1, 2 mg; vitamin B2, 4 mg; pantothenic acid, 15 mg; vitamin B6, 10 mg; vitamin B12, 0.05 mg; nicotinic acid, 30 mg; folic acid, 2 mg; vitamin K3, 1.5 mg; biotin, 0.2 mg; choline chloride, 800 mg; and vitamin C, 100 mg. The premix did not contain additional copper, zinc, antibiotics, or probiotics. ^2^ All data are the results of chemical analysis conducted in triplicate. CP: crude protein; DE: digestible energy.

**Table 2 animals-10-01998-t002:** Serum biochemical parameters of piglets ^1,2^.

Items	NC	PC	EUF	Pooled-SEM	*p*-Value
Day 15					
Total protein, g/L	48.29 ^b^	50.21 ^b^	59.82 ^a^	2.30	0.006
Albumin, g/L	32.15	35.94	37.45	3.04	0.552
Blood urea nitrogen, mmol/L	4.21	4.53	4.78	0.16	0.075
Glucose, mmol/L	6.78	6.49	6.53	0.20	0.629
Alanine transaminase, U/L	32.38 ^b^	38.59 ^ab^	47.37 ^a^	2.73	0.006
Aspartate aminotransferase, U/L	30.24	34.27	29.57	3.18	0.608
Alkaline phosphatase, U/L	214.24 ^ab^	199.27 ^b^	237.57 ^a^	9.20	0.029
Immunoglobulin G, g/L	1.36 ^b^	1.76 ^ab^	1.97 ^a^	0.15	0.036
Immunoglobulin M, g/L	0.36	0.45	0.48	0.04	0.121
Day 35					
Total protein, g/L	45.26 ^b^	49.67 ^ab^	52.98 ^a^	1.95	0.046
Albumin, g/L	28.56	29.79	32.97	3.21	0.631
Blood urea nitrogen, mmol/L	4.39	4.06	3.78	0.18	0.083
Glucose, mmol/L	6.87	6.79	6.94	0.16	0.838
Alanine transaminase, U/L	35.19	39.46	41.38	2.81	0.306
Aspartate aminotransferase, U/L	28.49	31.05	29.87	1.84	0.642
Alkaline phosphatase, U/L	197.56	214.56	227.69	9.33	0.104
Immunoglobulin G, g/L	1.27	1.58	1.87	0.18	0.088
Immunoglobulin M, g/L	0.29	0.32	0.28	0.03	0.710

^1^ NC = negative control, low-protein basal diet no antibiotics included; PC = positive control, low-protein basal diet + antibiotics (75 mg/kg quinocetone, 20 mg/kg virginomycin and 50 mg/kg aureomycin); EUF = *Eucommia ulmoides* flavones, low-protein basal diet + 0.01% EUF. ^2^ Values are the mean ± SEM, *n* = 8 per treatment group. ^a,b^ Mean values sharing different superscripts within a row differ (*p* < 0.05).

**Table 3 animals-10-01998-t003:** Lactic acid bacteria and coliform bacteria in the ileum and colon of piglets ^1,2^.

Log_10_ cfu/g		NC	PC	EUF	Pooled-SEM	*p*-Value
Day 15						
Lactic acid bacteria	Ileum	6.59 ^b^	7.35 ^ab^	7.49 ^a^	0.13	<0.001
	Colon	7.28	7.18	7.29	0.21	0.933
Coliform bacteria	Ileum	5.59 ^a^	4.58 ^b^	4.76 ^b^	0.19	0.003
	Colon	4.54	4.28	4.32	0.22	0.687
Day 35						
Lactic acid bacteria	Ileum	6.27	5.98	6.18	0.12	0.240
	Colon	7.15	6.87	7.09	0.25	0.721
Coliform bacteria	Ileum	4.79 ^a^	4.15 ^b^	4.06 ^b^	0.15	0.001
	Colon	4.35	4.28	4.21	0.13	0.751

^1^ NC = negative control, low-protein basal diet no antibiotics included; PC = positive control, low-protein basal diet + antibiotics (75 mg/kg quinocetone, 20 mg/kg virginomycin and 50 mg/kg aureomycin); EUF = *Eucommia ulmoides* flavones, low-protein basal diet + 0.01% EUF. ^2^ Values are the mean ± SEM, *n* = 8 per treatment group. ^a,b^ Mean values sharing different superscripts within a row differ (*p* < 0.05).

**Table 4 animals-10-01998-t004:** Jejunal and ileal morphology in piglets ^1,2^.

Items		NC	PC	EUF	Pooled-SEM	*p*-Value
Day 15						
Villus height, μm	Jejunum	246.47 ^b^	274.32 ^a^	276.35 ^a^	7.80	0.023
	Ileum	197.89 ^b^	241.32 ^a^	242.86 ^a^	12.79	0.035
Crypt depth, μm	Jejunum	105.62	89.57	102.45	7.85	0.348
	Ileum	86.57	74.54	78.54	7.22	0.551
Villus height: Crypt depth	Jejunum	2.33 ^b^	3.06 ^a^	2.70 ^ab^	0.16	0.020
	Ileum	2.29 ^b^	3.24 ^a^	3.09 ^ab^	0.24	0.028
Day 35						
Villus height, μm	Jejunum	197.23	226.89	234.35	12.37	0.151
	Ileum	168.54	187.43	198.48	14.91	0.362
Crypt depth, μm	Jejunum	112.54	105.84	104.58	7.94	0.777
	Ileum	98.57	99.45	97.58	4.13	0.951
Villus height: Crypt depth	Jejunum	1.75 ^b^	2.14 ^ab^	2.24 ^a^	0.11	0.017
	Ileum	1.71	1.89	2.03	0.20	0.542

^1^ NC = negative control, low-protein basal diet no antibiotics included; PC = positive control, low-protein basal diet + antibiotics (75 mg/kg quinocetone, 20 mg/kg virginomycin and 50 mg/kg aureomycin); EUF = *Eucommia ulmoides* flavones, low-protein basal diet + 0.01% EUF. ^2^ Values are the mean ± SEM, *n* = 8 per treatment group. ^a,b^ Mean values sharing different superscripts within a row differ (*p* < 0.05).

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
