# Peer review of "Eucommia ulmoides Flavones as Potential Alternatives to Antibiotic Growth Promoters in a Low-Protein Diet Improve Growth Performance and Intestinal Health in Weaning Piglets"

_animals, 2020, doi:10.3390/ani10111998_

Round 1

Reviewer 1 Report

Comments to the Authors of manuscript number: animals-960090 entitled “Eucommia Ulmoides Flavones as Potential Alternatives to Antibiotic Growth Promoters in a Low-protein Diet Improve Growth Performance and Intestinal Health in Weaning Piglets (Running title: Effect of Eucommia Ulmoides flavones on piglet 6 performance and gut health)”.

Chinese herbal products have been studied for many medical problems. Plant/herbal dietary supplements are studied in lots species. Many of them are not present elsewhere. Sometimes is difficult assess the possibility of herbal use. There is many questions such contamination with toxic compounds, heavy metals, pesticides of this plants.

  1. Authors should give date where the plant comes from. Short information about contamination.
  2. What is economic cost of introduction into breeding? breeding has its own market rights
  3. L63 avoid the term “ our group”
  4. Why low-protein diet was given should be written earlier in the part of “2.1”

5.L100 avoid the word of “killed”

6.L102- what part of intestine segment? It should be given clearly

7.part 2.3 – more information how it was analyzed should be added

8.part 2.4 were IgG and IgM analyzed as other biochemical parameters? Is it possible?

9.Table 2- all these parameters were in physiological range. It should be written and discused

  1. L248 – what is “reflect the body protein metabolic stasis”?

Author Response

Dear editor:

Thank you!

Jing

Reviewer 2 Report

Statistics:

  • Please, include further information related to the analysis of the time effect. How was it carried out? The authors stated that no differences were obtained between day 0 and 15 for some variables but the difference 15 - 0 seems different so I wonder if a tendency or diet x time interactions were observed.

Results:

  • Please, include some numeric results such as percentages of variation, differences… in order to describe the bar graphs and numbers from the tables more accurately and properly.

Discussion:

  • 90% of the discussion is a description of mechanisms. The authors should include more comparison and discussion with their own results.
  • Have the authors considered the possibility of combining antibiotics and flavones and check the results?

References:

  • The reference list shows 55 references. It is a big number considering such a short paper. Please, review the text, some of the citations seem to be redundant.

-Some English mistakes (grammar) can be detected. Please, correct.

Author Response

Dear editor:

Thank you!

Jing

Reviewer 3 Report

The manuscript entitled “Eucommia Ulmoides Flavones as Potential Alternatives to Antibiotic Growth Promoters in a Low-protein Diet Improve Growth Performance and Intestinal Health in Weaning Piglets” describes an alternative to the use of antibiotics free diets on swine production . The research has been well organised and the analysis of the results is complete. I think that the research might be very useful since alternative approach to the use of antibiotics must be find.

Line 158: Kruskal-Wallis is a not parametric test and it is used as an alternative to the parametric ANOVA one way. I think the authors may add “not” to the sentence “following by the Kruskal-Wallis test when data were NOT normally distributed”.

Line 202: Please separate “bacteria between” and “d 15”

Line 257: Please replace “defensing” with “defending”

Line 306: In the entire manuscript, the authors used two different form of the following word: “diarrhea or diarrhoea”. Please chose one.

I suggest adding a description of the Eucommia Ulmoides. I read is a small tree native in China, you could describe the importance of this plant in your country, its traditional use and if it could be easily available for the farmers. To know more information about the plant could help the scientific community to identify similar tools in each respective country.

Author Response

Dear editor:

Thank you!

Jing

Reviewer 4 Report

This paper explores the use of Eucommia Ulmoides Flavones in the diet as a means to replace antibiotics used in feed to prevent disease.

While there are some interesting findings I think it could be presented clearer. For example, you state that it improves growth and decreases diarrhea but it is actually the same as using antibiotics, which is the current commercial standard. This is still a great finding but I would change the angle so that you are comparing it with the normal feed that has antibiotics in it rather than the diet with no additives. This is also because you have used a low protein diet which I do not think is standard for weaners and therefore is not truly a control diet- what if this low protein diet was creating issues that the antibiotics and EUFs were able to manage but would not ordinarily be found? OR alternatively the low protein diet could be improving gut health and therefore the EUF and antibiotics have less challenges to deal with and are therefore more effective in this study than they would be commercially. Another concern that I have is that there are no day 0 samples and for the same reasons I am worried you are not truly measuring the changes that the diets are causing as you have no baseline to compare it to. They are still different across treatments which is a great finding, but how can you say villi length has been increased for EUF and antibiotic diets when in fact it could remain the same in these diets but be decreased in the low protein diet?

Some other considerations:

Line 61-63: The sentence about antibiotics seems out of place when this paragraph is about EUFs and we have already established that we need to eliminate antibiotics as they have been banned in feed, so it doesn't really matter if there is evidence to say they cause microbiome imbalance, especially since this is not brought up again.

Methods:

Is a low protein diet standard commercially?

Line 93: I don't understand why there are two phases when the nutritional levels are very similar. Can you confirm that this is due to slight changes in requirements for energy and protein and that the EUF and antibiotics remained the same across both phases?

Line 117: How did you measure feed intake? I am concerned that it appears it was only measured on two days in which case how can you be sure it is an accurate representation of their eating for each phase to calculate feed conversion ratios?

Line 162: Please confirm how you collected feed intake and calculated average daily feed intake.

Line 171: Figure 1B, isn't average daily gain showing the same information as 1a body weight?

Line 213: Figure 3, does fold of control mean fold change? Why have you presented it this way instead of using the actual values?

Line 237-239: in comparison to what?

Line 239: Wouldn't it be better to compare to normal protein not low protein? IF low protein has an effect on gut health then there is no true control. Also this could lead to less cases of diarrhea which would confound the effect of your EUF and antibiotic diets as they are more likely to reduce diarrhea if there is already only a low challenge.

There is some great stuff in this paper and very exciting results with the EUF addictive, I think you just need to work on better explaining the controls so that we can be sure the EUF is effective.

Author Response

Dear editor:

Thank you!

Jing

Round 2

Reviewer 1 Report

The manuscript can be published in this form.

Author Response

Dear Editor Phillips and Reviewers,

Thank you for your letter and for the reviewers’ comments concerning our manuscript titled “Eucommia Ulmoides Flavones as Potential Alternatives to Antibiotic Growth Promoters in a Low-protein Diet Improve Growth Performance and Intestinal Health in Weaning Piglets”. We were pleased to know that our work was rated as potentially acceptable for publication in Journal, subject to adequate revision.

Please review my itemized responses below and my yellow-highlighted revisions in the re-submitted files.

We hope that the revised manuscript is accepted for publication in the Animals.

Thanks again.

Best regards,

Jing

Reviewer# 1

The manuscript can be published in this form.

Thank you for agreeing to accept our manuscript.

Reviewer 2 Report

The authors have addressed the reviewers' comments properly.

Further description should be considered of the values characterized by statistical tendency from tables 2 and 3.

Author Response

Dear Editor Phillips and Reviewers,

Thank you for your letter and for the reviewers’ comments concerning our manuscript titled “Eucommia Ulmoides Flavones as Potential Alternatives to Antibiotic Growth Promoters in a Low-protein Diet Improve Growth Performance and Intestinal Health in Weaning Piglets”. We were pleased to know that our work was rated as potentially acceptable for publication in Journal, subject to adequate revision.

Please review my itemized responses below and my yellow-highlighted revisions in the re-submitted files.

We hope that the revised manuscript is accepted for publication in the Animals.

Thanks again.

Best regards,

Jing

Reviewer# 2

The authors have addressed the reviewers' comments properly.

Further description should be considered of the values characterized by statistical tendency from tables 2 and 3.

Thank you for the suggestion. We have revised line 197~198; 201~202; 213~214; 215~216. And we also added the description of the ALT result in the discussion section at line 277~282.
